# Machine learning augmented Tikhonov regularization with iterative approach for stable neutron spectrum unfolding

**Konstantin Chizhov**
Radiation safety department
Joint Institute for Nuclear Research,
Dubna, Russia.
University "Dubna",
Dubna, Russia.
kchizhov@jinr.ru

**Stepan Shirkov**
Radiation safety department
Joint Institute for Nuclear Research,
Dubna, Russia.
stepanshirkov@jinr.ru

## ABSTRACT

A hybrid multi-stage algorithm is developed for solving the ill-posed inverse problem of unfolding the neutron energy spectrum from multi-sphere Bonner spectrometer measurements. Traditional approaches, such as Tikhonov regularization and iterative methods, have significant limitations due to the subjective choice of the regularization parameter or initial approximation, which compromises the solution's stability and accuracy. In the proposed method, the first stage automated machine learning (autoML) is used to find the optimal model to predict the global spectral shape. The second stage applies Tikhonov regularization, where regularization parameter is objectively optimized based on a similarity metric relative to the autoML prediction. The smoothing functional is minimized using convex optimization techniques. The third stage utilizes the obtained solution as the initial guess for an iterative refinement procedure. Physical prior knowledge is incorporated both through a parametrically generated training dataset (weighted sums of fission, evaporation, Gaussian, and high-energy spectral components). The hybrid approach demonstrates better robustness to noisy input data compared to methods using solely Tikhonov regularization or machine learning. The developed methodology is applicable to neutron dosimetry at high-energy nuclear facilities and for solving a broad class of inverse problems described by Fredholm integral equations of the first kind.

## 1 INTRODUCTION

### 1.1 PROBLEM RELEVANCE

Accurate characterization of a wide range neutron energy spectra from eV to hundreds of MeV is necessary for radiation protection, shielding design and dosimetry for personnel of particle accelerators. Neutrons (as uncharged particles) require indirect detection methods. The Bonner Sphere Spectrometer (BSS) as a thermal neutron detector surrounded by polyethylene moderators of varying diameters, is the most widely used instrument for neutron spectrometry (Chizhov et al., 2025a). However, calculating the neutron energy spectrum $\phi(E)$ from the set of BSS readings constitutes an ill-posed inverse problem, described by a system of Fredholm integral equation of the $1^{st}$ kind. This ill-posedness, characterized by non-uniqueness and high sensitivity to measurement noise, necessitates specialized unfolding techniques.

### 1.2 PROBLEM STATEMENT

The relationship between the neutron spectrum $\phi$ and the BSS readings for $M$ spheres is given by:

$$Q_j = \int_{E_{\min}}^{E_{\max}} R_j(E)\,\phi(E)\,dE, \quad j = 1, \ldots, M, \tag{1}$$

where $R_j(E)$ is the response function of the $j$-th sphere, (Com, 2001). $Q_j$ is the Bonner spectrometer reading including measurement error $\epsilon$, $E$ is the neutron energy, $E_{min} = 10^{-9} MeV$ and $E_{max} = 631 MeV$.

Since the unfolding of the neutron spectrum is assumed to be a fairly wide energy range that exceeds 11 orders of magnitude, to reduce the error of numerical integration it is convenient to convert it to a new variable called lethargy $u(E) = \lg(E/E_{\min})$:

$$\ln 10 \times \int_0^{l_E} K_j(u)\,\phi(u)\,E(u)\,du = Q_j, \quad j = 1, \ldots, M, \tag{2}$$

where $l_E = \lg(E_{\max}/E_{\min})$.

The integrand form suggests finding $\varphi(u) = \phi(u) \cdot E(u)$, which we continue to call the neutron spectrum.

Discretizing the energy into $N$ bins ($N \gg M$) transforms equation 2 into a system of linear equations:

$$\mathbf{A}\boldsymbol{\varphi} = \mathbf{q}, \tag{3}$$

where $\mathbf{A} \in \mathbb{R}^{M \times N}$ is the response matrix, $\boldsymbol{\varphi} \in \mathbb{R}^N$ is the discretized fluence and $\mathbf{q} = \{Q_1, ..., Q_M\} \in \mathbb{R}^M$ is the vector of measured counts. The system is underdetermined and ill-conditioned.

This inverse problem can be reformulated as a machine learning regression task. The input features are the $M$ normalized BSS readings $\tilde{\mathbf{q}} = \mathbf{q}/k$, where $k = \sum_{j=1}^{M} Q_j$. The target output is the discretized spectrum vector $\boldsymbol{\varphi}$. The ML model $\mathcal{F}$ learns the mapping:

$$\boldsymbol{\varphi}_{\text{ML}} = \boldsymbol{k}\mathcal{F}(\tilde{\mathbf{q}}). \tag{4}$$

## 1.3 Existing Approaches and Related Work

Neutron spectra unfolding methods include iterative algorithms (e.g., MLEM (Adler et al., 2018), Landweber (Chizhov & Chizhov, 2025)), maximum entropy deconvolution (MAXED) (Borshchev et al., 2026), Tikhonov regularization (Chizhov & Chizhov, 2024), truncated singular value decomposition (TSVD) (Chizhov & Chizhov, 2025), stochastic methods like genetic algorithms (Freeman et al., 1999), parametric (Bedogni et al., 2007; Sannikov et al., 2007) and other methods (Gómez-Ros et al., 2022; Chizhov et al., 2026a; Machado et al., 2023; Bouchama et al., 2024). However, the results of application of a method depends on the chosen parameters. For iterative methods, the initial approximation and the choice of the number of iterations are important. For regularization methods, the choice of the regularization parameter can over smooth the solution or create nonphysical ripples in the spectrum if the parameter is too small. In unfolding process it is necessary to consider the physical correctness of the spectrum: smoothness and non-negativity. Conditional stability of the solution is needed, otherwise small errors in BSS readings may produce large uncertainty in spectra.

Machine learning (Chizhov, 2025) and neural networks (NN) (Ortiz-Rodriguez et al., 2014) of different types and architecture has been increasingly applied to this problem, including deep learning frameworks with input feature transformations (Chizhov & Bely, 2025), Kolmogorov-Arnold networks (Starikovskaya & Chizhov, 2026), radial basis function (Alvar et al., 2017), convolutional (Bouhadida et al., 2023), generalized regression (del Rosario Martinez-Blanco et al., 2016) and Bayesian NN (Zhou et al., 2025). For interpretation of the NN spectrum prediction, explainable artificial intelligence (XAI) methods such as SHAP (Chizhov, 2025), LIME (Chizhov et al., 2024), ANFIS-based (Chizhov et al., 2026b) were applied.

ML models are typically trained on synthetic datasets generated via Monte Carlo codes (Bouhadida et al., 2023) or parametric models (McGreivy et al., 2023). A set of real spectra may also be incorporated into the dataset (Chizhov, 2025) with assumption that these spectra have been accurately unfolded via the methods employed by the authors. These datasets pair a neutron spectra with corresponding BSS responses.

Finding the optimal model requires iterating over both the algorithms themselves and the model's hyperparameters, as well as consuming a significant amount of computational time. automatic ML AutoML frameworks (Chizhov et al., 2025b), which include algorithms for efficiently finding hyperparameters within a given time, simplify this process.

However, data-driven ML approaches can struggle with generalization to spectra outside the training distribution and may lack physical consistency, for example producing negative fluence values or being overly sensitive to input noise. Therefore, authors propose in this work a synergistic approach: an ML model provides a rapid, data-driven initial guess, which is then refined by well-established algorithms, combining the speed of ML with the stability and physical and mathematical correctness, as it was made in (Shiwei et al., 2024) for the dataset of 201 neutron energy spectra and the corresponding measured responses from (Com, 2001) .

In this paper we propose a method with combination of algorithms, where on the first stage autoML trained on a large synthetic dataset is used to find the optimal model to predict the global spectral shape. This shape was the init spectra for next method. For Tikhonov regularization unfolding, regularization parameter is objectively optimized based on a similarity metric relative to the autoML prediction. The smoothing functional is minimized using convex optimization techniques (Diamond & Boyd, 2016). For iterative methods, the unfolded by other method spectra was the initial guess. The uncertainty in spectrum unfolding was estimated using a Monte Carlo simulation for a set of random samples and a normally distributed error. Combination of methods yields a robust solution for the noisy initial data.

## 2 Materials and Methods

### 2.1 Synthetic Dataset Generation

A dataset of $5 \times 10^5$ neutron spectra was generated. Each spectrum $\varphi(E)$ is modeled as a linear combination of four parametric functions (Starikovskaya & Chizhov, 2025) representing thermal, epithermal, fast, and high-energy components. The energy range was discretized into $N = 60$ bins in logarithmic scale.

For each spectrum, the readings $\mathbf{q}$ for a 10 sphere BSS GSF (Com, 2001) (with diameters of 2, 3, 5, 6, 8, 10, 12, 15, 18 inches and bare detector without the moderator sphere) were calculated by convolving the spectrum with response functions $\mathbf{A}$. The dataset was split into training (70%) and testing (30%) sets.

### 2.2 Machine Learning Model and Feature Processing

In this work we used H2O (LeDell et al., 2020) – a ML framework for algorithm selection, feature generation, hyperparameter tuning, iterative modeling and model assessment. The H2O AutoML uses a combination of fast random search and stacked ensembles to achieve the best model metrics. H2O AutoML includes XGBoost, gradient boosting machines (GBM), random forests, deep NN and generalized linear models (GLM).

We trainted $N = 60$ independent models for each energy bin. Each was trained to predict the normalized spectrum $\tilde{\varphi}$ from the normalized BSS readings $\tilde{\mathbf{q}}$. Model performance was evaluated using the coefficient of determination ($R^2$) and root mean squared error $RMSE$. A post-processing step sets all negative values in $\varphi_{\mathrm{ML}}$ to zero to ensure physical correctness before it is used as an initial guess for iterative methods. Training time was limited to 300 seconds for each model.

### 2.3 Refinement Algorithms

#### 2.3.1 Tikhonov Regularization

The solution for equation 3 is found by minimizing the regularized functional, equation 5:

$$\varphi^* = \arg\min_{\varphi \geq 0} \left\{ \|\mathbf{A}\varphi - \mathbf{q}\|_2^2 + \alpha\|\mathbf{L}\varphi\|_2^2 \right\}, \tag{5}$$

where $\mathbf{L}$ is the identity matrix (zeroth-order Tikhonov) and $\alpha \geq 0$ is the regularization parameter. The parameter $\alpha$ is chosen to maximize the similarity metric between the Tikhonov solution and the ML estimate $\varphi_{\mathrm{ML}}$. In this work a cosine similarity (CS) (Condon, 2024) was chosen as the metric (equation 6). It is a measure of similarity between two non-zero vectors in n-dimensional space. For spectra the values of the vector cannot be negative, so for our task the value of CS is bounded in [0,1]. Two proportional vectors (spectra) have a CS of 1, for two orthogonal vectors CS = 0.

$$\alpha^* = \arg\max_{\alpha} \frac{\langle \boldsymbol{\varphi}(\alpha), \boldsymbol{\varphi}_{\mathrm{ML}} \rangle}{\|\boldsymbol{\varphi}(\alpha)\| \|\boldsymbol{\varphi}_{\mathrm{ML}}\|}. \tag{6}$$

The algorithm was developed as a python code with *CVXPY* (Diamond & Boyd, 2016) package and ECOS (Domahidi et al., 2013) convex solver.

### 2.3.2 MAXIMUM LIKELIHOOD EXPECTATION MAXIMIZATION

Maximum likelihood expectation maximization (MLEM) is an iterative process that maximizes the likelihood of obtaining the measured data when convergence is achieved, providing an accurate neutron spectrum (Díaz-Comeche et al., 2025). The MLEM algorithm iteratively updates the spectrum estimate for $k = 3000$ steps:

$$\varphi_i^{(k+1)} = \frac{\varphi_i^{(k)}}{\sum_{j=1}^{M} A_{ji}} \sum_{j=1}^{M} \frac{q_j A_{ji}}{\sum_{l=1}^{N} A_{jl}\varphi_l^{(k)}}, \quad i = 1, \dots, N, \tag{7}$$

starting from the ML estimate $\boldsymbol{\varphi}^{(0)} = \boldsymbol{\varphi}_{\mathrm{ML}}$.

The algorithm was developed as a python code with *ODL* (Adler et al., 2018) package.

### 2.3.3 LANDWEBER ITERATION

The Landweber iteration is a gradient descent method for solving $\mathbf{A}\boldsymbol{\varphi} = \mathbf{q}$:

$$\boldsymbol{\varphi}^{(k+1)} = \boldsymbol{\varphi}^{(k)} + \alpha \mathbf{A}^T (\mathbf{q} - \mathbf{A}\boldsymbol{\varphi}^{(k)}), \tag{8}$$

with relaxation parameter $\alpha$ chosen as $\alpha = 1/\|\mathbf{A}^T\mathbf{A}\|_2$, and initialized with $\boldsymbol{\varphi}^{(0)} = \boldsymbol{\varphi}_{\mathrm{ML}}$, for chosen $k = 3000$ steps:.

These unfolding methods were implemented to the python package *bssunfold* (Chizhov, 2026).

### 2.4 ROBUSTNESS EVALUATION

To assess stability against measurement uncertainties, random Gaussian noise with a relative standard deviation of $\zeta_Q = 1\%$) was added to the simulated detector readings $\mathbf{q}$. For $N_{random} = 500$ random samples of readings spectra were unfolded by methods and by mixture of methods.

### 2.5 EFFECTIVE DOSE RATE ASSESSMENT

Ensuring the health and safety of personnel working at nuclear and accelerator facilities is a fundamental aspect of radiation protection. The primary objective of radiation safety is to protect the health of the public, including occupationally exposed individuals, from the harmful effects of ionizing radiation through the implementation of established principles and regulatory standards. In accordance with radiation safety norms (NRB, 2009), exposure of personnel to neutrons must be below specified limits. To evaluate and compare the quality of different unfolding procedures, the effective dose rate for isotropic irradiation (ISO), denoted as $\dot{H}$, was used as the metric. This quantity $\dot{H}$ is calculated by integrating the product of the neutron fluence spectrum $\phi(E)$ and the corresponding fluence to dose conversion coefficient $h(E)$ for monoenergetic particles in a certain irradiation geometry (Com, 2001; Petoussi-Henss & et al, 2010) over the energy range of interest, as expressed in equation 9:

$$\dot{H} = \int_{E_{min}}^{E_{max}} \phi(E) \cdot h(E) \, dE. \tag{9}$$

### 2.6 SPECTRA SIMILARITY METRICS

To assess the quality of spectrum unfolding, we used metrics such as MSE, Wasserstein distance (WD), CS, relative difference in the effective dose rate $\Delta\dot{H}$ and the residual norm for effective and reference BSS readings ($\Delta\mathbf{q}$) (Chizhov et al., 2025b). This set of metrics allows us to evaluate the unfolding quality for both individual energy bins and the spectrum shape as a whole.

## 3 RESULTS

We have considered the following unfolding methods and its combinations: H2O, CVXPY with ECOS solver, Landweber, MLEM, H2O-CVXPY, H2O-MLEM, H2O-MLEM-CVXPY, H2O-Landweber,H2O-CVXPY-Landweber, H2O-MLEM-CVXPY-Landweber, H2O-CVXPY-MLEM. Algorithms were validated for GSF BSS (Com, 2001) on reference spectra $^{252}Cf$ and the Monte-Carlo calculated spectra from GSF realistic neutron field facility (GRENF), for position C that corresponds to a specific physical placement of the dosimeter relative to the plutonium source with significant scattering from surrounding materials (Com, 2001).

### 3.1 ML MODEL PERFORMANCE

Models were trained on the HybriLIT platform (Anikina et al., 2024). It achieved an accuracy comparable to the values we had previously obtained using other ML algorithms (Chizhov & Bely, 2025; Chizhov, 2025), mean $R^2$ on training set is 0.92 and mean $R^2$ on test set – 0.86. H2O autoML framework found XGBoost and GBM as the best algorithms with individual hyperparameters for each model.

### 3.2 RESULTS FOR $^{252}Cf$ SPECTRA

This spectrum is good for validation. It can be measured experimentally using other methods or calculated using Monte Carlo programs. It has one peak, and the rest of the neutron energy range is zero. If the unfolding method produces oscillations near zero or negative values, this will be visible on the spectrum.

The ML model unfolded the spectrum quite well ($CS = 0.96$), correctly identifying the peak, although slightly undershooting it. There are some minor artifacts on the right side of the spectrum. Effective BSS readings were calculated by equation 3 and are close to real measurements, Fig. 1.

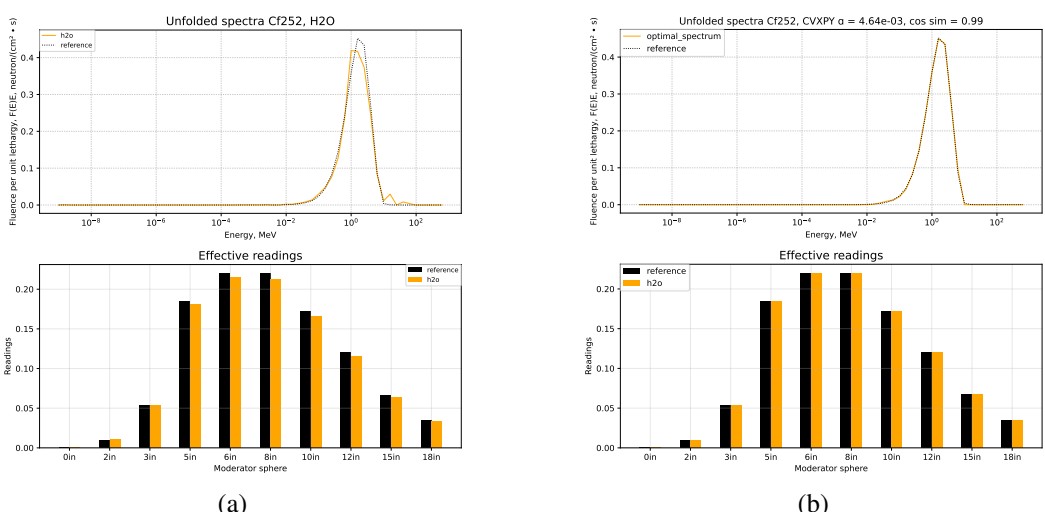

Figure 1: (a) - reference and unfolded by H2O autoML spectra for $^{252}Cf$ and effective BSS readings, (b) - Tikhonov regularization with parameter selected according to the best CS with H2O spectrum.

For $^{252}Cf$ MLEM and Landweber iterative algorithms with an uniform spectrum as the initial approximation yield better results than the ML, with slightly lower peak heights, the CS for these methods is 0.99. With the correct selection of the regularization parameter, Tikhonov's method gives the best result, a near-perfect match ($CS = 0.999$), Fig. 3.2. But if we use use a combination of methods, then the initial approximation from ML allows both the iterative and regularization methods to obtain excellent agreement with reference spectra, Fig. 1, 3.2.

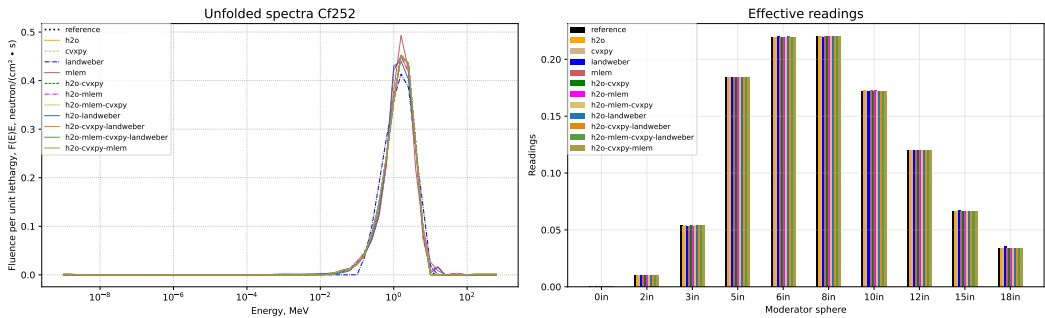

Figure 2: Comparison of reference and unfolded spectra and readings for $^{252}Cf$.

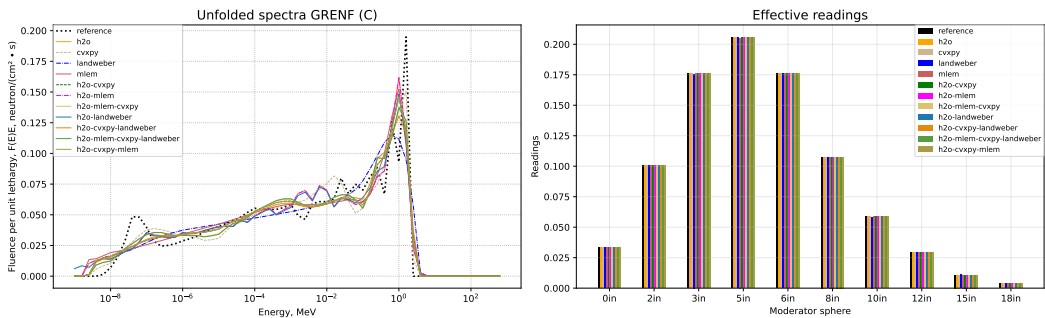

Figure 3: Reference and unfolded spectra for GRENF, position C.

### 3.3 RESULTS FOR GRENF, POSITION C SPECTRA

This spectra has a wide neutron energy and it has a sharp peak, that is difficult to unfold.

Although the algorithms selected spectra with almost ideal values of effective readings, the shape of the spectra themselves is different, Fig. 3. CS for H2O is 0.95 and $R^2 = 0.79$, H2O with CVXPY has $CS = 0.96$ and $R^2 = 0.83$, the pipeline of H2O with MLEM and CVXPY gives $CS = 0.97$ and $R^2 = 0.86$.

### 3.4 STABILITY ANALYSIS

Measurement results always contain errors. In ill-posed problems, small errors in the input data can lead to large errors in the solution. Therefore, we proposed combination of methods to stabilize the solution. Spectra were unfolded for input noise level $\zeta_Q$ and $N_{random}$ samples. For evaluation metrics from section 2.6 we obtained results for GRENF(C) presented in table 1.

For the GRENF(C) spectrum Figure 4 shows that the spectrum shape changes with errors in the source data: the range of possible spectra is shown in the semi-transparent area. Iterative algorithms, which converge to a spectrum similar to the true one despite the error, prove to be the most stable. Tikhonov's method produces excellent results with accurate data, but small errors significantly alter the spectrum shape. To achieve conditional stability of the solution, we propose to use iterative methods in the pipeline. This yields a more accurate solution and reduces the uncertainty range. A combination of methods allows for the highest accuracy with an acceptable error. Selecting a regularization parameter based on the spectrum shape of the reconstructed ML yields significantly better results than using Tikhonov's method alone. The best result is $R^2 = 0.76$, MSE $= 3.1 \times 10^{-4}$ for both H2O-CVXPY and MLEM method.

Table 1: GRENF(C). Average metrics with standard deviation for assessing the similarity of a spectrum to a reference one for various unfolding methods taking into account errors in input data.

| Method | $R^2$ | CS | MSE |
|---|---|---|---|
| h2o | $0.69 \pm 0.09$ | $0.93 \pm 0.02$ | $(4 \pm 1.2) \times 10^{-4}$ |
| cvxpy | $-5.03 \pm 4.15$ | $0.55 \pm 0.16$ | $(7.9 \pm 5.4) \times 10^{-3}$ |
| landweber | $0.75 \pm 0.02$ | $0.94 \pm 0.01$ | $(3.3 \pm 0.3) \times 10^{-4}$ |
| mlem | $0.76 \pm 0.03$ | $0.94 \pm 0.01$ | $(3.1 \pm 0.4) \times 10^{-4}$ |
| h2o-cvxpy | $0.76 \pm 0.04$ | $0.94 \pm 0.01$ | $(3.1 \pm 0.5) \times 10^{-4}$ |
| h2o-mlem | $0.69 \pm 0.09$ | $0.93 \pm 0.02$ | $(4 \pm 1.2) \times 10^{-4}$ |
| h2o-mlem-cvxpy | $0.73 \pm 0.08$ | $0.94 \pm 0.02$ | $(3.5 \pm 1.1) \times 10^{-4}$ |
| h2o-landweber | $0.69 \pm 0.09$ | $0.93 \pm 0.02$ | $(4 \pm 1.2) \times 10^{-4}$ |
| h2o-cvxpy-landweber | $0.73 \pm 0.09$ | $0.94 \pm 0.02$ | $(3.5 \pm 1.2) \times 10^{-4}$ |
| h2o-mlem-cvxpy-landweber | $0.69 \pm 0.13$ | $0.93 \pm 0.03$ | $(4 \pm 1.7) \times 10^{-4}$ |
| h2o-cvxpy-mlem | $0.73 \pm 0.08$ | $0.93 \pm 0.02$ | $(3.6 \pm 1.2) \times 10^{-4}$ |
| Method | WD | $\Delta \dot{H}, \%$ | $\Delta \mathbf{q}$ |
| h2o | $0.63 \pm 0.2$ | $1.16$ | $2.7 \times 10^{-3}$ |
| cvxpy | $2.33 \pm 0.75$ | $3.8$ | $3.4 \times 10^{-3}$ |
| landweber | $0.48 \pm 0.15$ | $1.2$ | $2.8 \times 10^{-3}$ |
| mlem | $0.52 \pm 0.12$ | $0.97$ | $2.6 \times 10^{-3}$ |
| h2o-cvxpy | $0.52 \pm 0.16$ | $1.18$ | $2.8 \times 10^{-3}$ |
| h2o-mlem | $0.63 \pm 0.2$ | $1.16$ | $2.7 \times 10^{-3}$ |
| h2o-mlem-cvxpy | $0.68 \pm 0.24$ | $1.41$ | $2.9 \times 10^{-3}$ |
| h2o-landweber | $0.74 \pm 0.25$ | $1.49$ | $2.9 \times 10^{-3}$ |
| h2o-cvxpy-landweber | $0.7 \pm 0.27$ | $1.53$ | $2.9 \times 10^{-3}$ |
| h2o-mlem-cvxpy-landweber | $0.81 \pm 0.32$ | $1.63$ | $3 \times 10^{-3}$ |
| h2o-cvxpy-mlem | $0.66 \pm 0.27$ | $1.3$ | $2.6 \times 10^{-3}$ |

## 4 DISCUSSION

The results demonstrate that the proposed hybrid framework successfully leverages the complementary strengths of ML and analytical methods. The ML model provides a physically reasonable starting point that is already close to the true solution, effectively narrowing the solution space. This prior information is used in regularization and iterative algorithms and it gives better results for MLEM and Landweber algorithms, if spectra has a complex form with sharp peaks.

The strategy of selecting the Tikhonov parameter $\alpha$ by maximizing CS with the ML prior is a data-driven alternative to classical methods like the L-curve (Koslowsky, 2020). It automates parameter tuning and adapts it to the specific spectral shape suggested by the measurements, leading to more consistent performance. Of course, the result depends heavily on the trained model. It may fail to capture sharp peaks, which can be identified and unfolded more accurately with manual parameter selection (in accordance with the expert knowledge of spectra shape) and another solver such as Gurobi, Express (Chizhov et al., 2026a; Diamond & Boyd, 2016). But if we don't know the shape, autoML selection can significantly improve the unfolding. It is possible to use the WD or maximum mean discrepancy as a metric of spectral similarity and regularization parameter selection (Chizhov et al., 2025b).

The primary limitation remains the dependence of the ML model on the quality and representativeness of the training data. Spectra very far from the FRUIT-generated distribution (Bedogni et al., 2007) may lead to poor initial guesses. The model metrics can be improved by incorporating a broader set of spectra from experimental and particle transport simulation software into the training dataset. Using combination of other types of ML algorithms and NN architectures with specific tabular data transformation (Thielmann et al., 2024; Chizhov & Bely, 2025) can also yield better results.

Addition in the unfolding chain methods, such as variations of the conjugate gradient and quasi-Newton methods (e.g. L-BFGS-B (Zhu et al., 1997; Diamond & Boyd, 2016)) could also improve

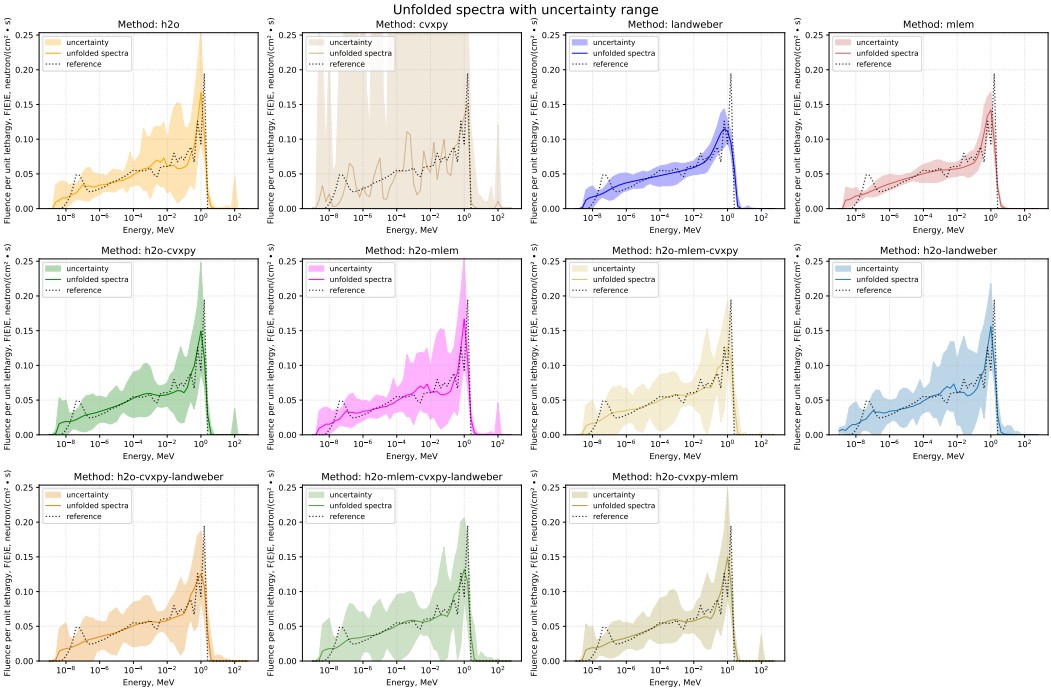

Figure 4: GRENF(C). Average unfolded spectra for developed unfolding algorithms, semi-transparent area shows the uncertainty of unfolding.

the unfolding results since they can give more accurate results than the methods used in this work (Chizhov et al., 2026a).

The selection of the optimal set of moderator spheres is important for spectrum unfolding accuracy. It is necessary to ensure complete coverage of the neutron energy range. It is advisable to use spheres with the most diverse response functions. In practice, this means avoiding the use of multiple spheres of similar diameters. This reduces the correlation between the rows of the response matrix and lowers matrix's condition number (Chizhov & Chizhov, a;b). The uncertainty of response functions for moderator spheres is not taken into account in calculations, the estimates for it are given in (Cao et al., 2015), for neutron energies above about 20 MeV in (Pioch et al., 2010).

On a regular basis, dose assessment error must achieve within $30\%$ for neutron dose measurements (Condon et al., 2025). The proposed pipeline for $\zeta_Q = 1\%$ gives the error in estimation of the $\dot{H}$ from the spectrum around $1.5\%$. For $\zeta_Q = 5\%$ noise in readings, as is usually the case with BSS measurements, the pipeline gives around $9\%$ $\dot{H}$ uncertainty, which is less than the recommended threshold.

### 4.1 MODEL INFERENCE

The most computer time is spent on training the model, since it needs to be done for each set of moderator spheres and for each BSS. The hybrid multi-stage algorithm works quickly even for a trained model, allowing for the neutron spectra unfolding and assessment of doses to personnel. It is implemented as a python code for cross-platform and convenient use by radiation safety specialists.

To facilitate rapid inference, the trained in H2O framework ML models were serialized into the Open Neural Network Exchange (ONNX) format (Panchal et al., 2023). The conversion process from the H2O framework utilized the MOJO (Model Object, Optimized) format as an intermediate step. But used version of *onnxmltools* (the MOJO to ONNX conversion tool) lacks support for XGBoost models. For our models only 13 were XGBoost, while the remaining were GBM. To maintain workflow integrity while maximizing the benefits of ONNX optimization, a specialized model pipeline was developed, incorporating only the GBM models.

This representation ensures interoperability, allowing the models to be deployed and executed across a diverse range of platforms and programming languages, including Python, Java, JavaScript, mobile environments and C/C++. Furthermore, the runtime efficiently leverages hardware accelerators. Consequently, a model executed within the ONNX Runtime framework can substantially outperform its counterpart running in the original training framework, particularly when deployed with the C++ based tools (Sengupta & Moneta, 2025). Translation of ONNX to Field Programmable Gate Arrays (FPGA) is also supported (Manca et al., 2024).

The developed Python unfolding algorithms (MLEM, Landweber iteration, Tikhonov regularization) can be rewritten in C or compiled to machine code using Cython or Numba JIT (Just-In-Time) (Lam et al., 2015). The ECOS (Domahidi et al., 2013) solver was originally written as library-free ANSI-C code. Then it can be synthesized for FPGA via High-Level Synthesis (HLS) tools (Huang et al., 2021), or directly from Python using specialized frameworks (Uguen & Petit, 2018; Huang et al., 2021). Integrating these algorithms into BSS hardware as a System-on-Chip (SoC) is promising: FPGAs offer hardware acceleration, execution flexibility, and energy efficiency for real-time spectrum unfolding.

## 5 CONCLUSIONS

A hybrid multi-stage algorithm for neutron spectrum unfolding has been developed, combining a machine-learned prior with Tikhonov regularization and iterative algorithms (Landweber, MLEM). The algorithm uses an set of automatically fitted in H2O framework models (XGboost, GBM) trained on a large synthetic dataset to generate an initial spectrum estimate. This estimate is post-processed for physical consistency and then used to initialize the Tikhonov regularization process with the regularization parameter selected via optimization of cosine similarity.

The proposed approach was validated on a test set of spectra derived from Monte Carlo simulations: spectra for $^{252}Cf$ and realistic neutron field facility (GRENF). The stability against measurement noise compared to standalone ML or traditional unfolding algorithms was accessed.

It has been shown that the combination of a machine learning algorithm with a regularization method and the iteration algorithm provides high spectrum unfolding accuracy while reducing uncertainty caused by measurement errors. For example, with a measurement error of $1\%$, the error in estimation of the effective dose rate from the spectrum is $1.5\%$, for measurement error of $5\%$ it is $9\%$.

In terms of the residual norm, the method yields effective reading values for BSS very close to the actual measurement data. For GRENF(C) spectra the machine learning model has $R^2 = 0.69$, while the combined algorithm gives $R^2 = 0.76$.

This work establishes a practical and effective pipeline for spectra unfolding with machine learning and inverse problem solvers. The proposed method could be used for improving radiation protection in high-energy neutron fields.

### ACKNOWLEDGMENTS

Authors acknowledge the support of the JINR Multifunctional Information and Computing Complex for providing computational resources, http://hlit.jinr.ru.

### FUNDING

This work was carried out within the framework of the state assignment of the Ministry of Education and Science of the Russian Federation (project No. 124112200072-2, "Application of Explainable Artificial Intelligence for the Interpretation of Machine Learning Algorithms").

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
