# OpenReview forum: "Machine learning augmented Tikhonov regularization with iterative approach for stable neutron spectrum unfolding"
_mathai.club/MathAI/2026/Conference — 2026 Oral_

### Official Review · Reviewer_9Qs2 · 2026-03-11
**Machine learning augmented Tikhonov regularization with iterative approach**

**Rating:** 6
**Confidence:** 4

**Review:**

This paper addresses a hybrid multi-stage algorithm for neutron spectrum unfolding has been developed, combining a machine-learned prior with Tikhonov regularization and iterative algorithms.
Pros:
1. A significant set of different algorithms and their combinations are considered.
2. The paper is formatted according to the conference template.
Cons:
1. In line 71, in the formula for k[j] (j is the index), there is an index j to the left of the equal sign, but not to the right.
2. The results of applying various algorithm combinations to the realistic  spectra GRENF (Table 1) almost always show little difference.
The determination coefficient lies in a narrow range from 0.69 to 0.76—seemingly good, but far from ideal. There are no clear recommendations for using any particular combination of the selected algorithms.
3. The article does not discuss the problems of deep machine learning.

---

> ### Author Rebuttal · Authors · 2026-03-13
>
> Thanks for your comments, it will be taken into account when finalizing the article.
> Deep learning for this topic is discussed in paper: Chizhov K, Bely A. Neutron spectrum unfolding using deep learning models for tabular data, Moscow University Physics Bulletin, doi 10.3103/S0027134925702832 that is in references

---

### Decision · Program_Chairs · 2026-03-14

**Decision:**

Accept (Oral)

**Comment:**

Dear Author(s),

On behalf of the Program Committee of the International Conference on Mathematics of Artificial Intelligence (MathAI 2026), we are pleased to inform you that your paper has been accepted for an oral presentation at MathAI 2026.

Your paper was evaluated through a rigorous two-stage review process involving both automated screening and expert review by members of the Program Committee. The reviewers recognized the quality and contribution of your work.

Presentation details:

- Format: Oral presentation (15–20 minutes + 5 minutes Q&A)
- Mode: You may present either in person (offline) at the conference venue in Sirius, Russia, or remotely via Zoom. Please indicate your preferred mode when confirming your participation.
- Conference dates: Marh 30 - April 3, 2026
- Website: https://mathai.club

Next steps:

1. Please confirm your participation and presentation mode by replying to this email mathai.club@yandex.ru no later than March 15, 2026 18:00 Moscow time.
2. If you plan to attend in person, the organizing committee will provide accommodation details separately.
3. Please prepare your final camera-ready manuscript according to the formatting guidelines available at https://mathai.club and upload it to OpenReview by March 15, 2026 18:00 Moscow time.

Should you have any questions regarding the program, logistics, or your presentation slot, please do not hesitate to contact us.

We look forward to your contribution to MathAI 2026.

With kind regards,

MathAI 2026 Program Committee
International Conference on Mathematics of Artificial Intelligence
https://mathai.club
OpenReview: https://openreview.net/group?id=mathai.club/MathAI/2026/Conference
Telegram: https://t.me/MathAI_club
Email: mathai.club@yandex.ru